# Risk Stratification of Patients with Pulmonary Arterial Hypertension: The Role of Echocardiography

**DOI:** 10.3390/jcm11144034

**Published:** 2022-07-12

**Authors:** Valentina Mercurio, Hussein J. Hassan, Mario Naranjo, Alessandra Cuomo, Jeremy A. Mazurek, Paul R. Forfia, Aparna Balasubramanian, Catherine E. Simpson, Rachel L. Damico, Todd M. Kolb, Stephen C. Mathai, Steven Hsu, Monica Mukherjee, Paul M. Hassoun

**Affiliations:** 1Division of Pulmonary and Critical Care Medicine, Department of Medicine, Johns Hopkins University, Baltimore, MD 21287, USA; hhassan6@jhmi.edu (H.J.H.); mnaranj7@jhmi.edu (M.N.); abalasu2@jhmi.edu (A.B.); catherine.simpson@jhmi.edu (C.E.S.); rdamico1@jhmi.edu (R.L.D.); toddkolb@jhmi.edu (T.M.K.); smathai4@jhmi.edu (S.C.M.); 2Department of Translational Medical Sciences, Federico II University, 80131 Naples, Italy; alebcuomo@gmail.com; 3Cardiovascular Division, Department of Medicine, Perelman School of Medicine, University of Pennsylvania, Philadelphia, PA 19104, USA; jeremy.mazurek@pennmedicine.upenn.edu; 4Pulmonary Hypertension, Right Heart Failure and Pulmonary Thromboendarterectomy Program, Temple University Hospital, Philadelphia, PA 19140, USA; paul.forfia@tuhs.temple.edu; 5Division of Cardiology, Department of Medicine, Johns Hopkins University, Baltimore, MD 21287, USA; steven.hsu@jhmi.edu (S.H.); mmukher2@jhu.edu (M.M.)

**Keywords:** pulmonary arterial hypertension, echocardiography, risk stratification, survival

## Abstract

Background: Given the morbidity and mortality associated with pulmonary arterial hypertension (PAH), risk stratification approaches that guide therapeutic management have been previously employed. However, most patients remain in the intermediate-risk category despite initial therapy. Herein, we sought to determine whether echocardiographic parameters could improve the risk stratification of intermediate-risk patients. Methods: Prevalent PAH patients previously enrolled in observational studies at 3 pulmonary hypertension centers were included in this study. A validated PAH risk stratification approach was used to stratify patients into low-, intermediate-, and high-risk groups. Right ventricular echocardiographic parameters were used to further stratify intermediate-risk patients into intermediate-low- and intermediate-high-risk groups based on transplant-free survival. Results: From a total of 146 patients included in our study, 38 patients died over a median follow-up of 2.5 years. Patients with intermediate-/high-risk had worse echocardiographic parameters. Tricuspid annular plane systolic excursion (TAPSE) and the degree of tricuspid regurgitation (TR) were highly associated with survival (*p* < 0.01, *p* = 0.04, respectively) and were subsequently used to further stratify intermediate-risk patients. Among intermediate-risk patients, survival was worse for patients with TAPSE < 19 mm compared to those with TAPSE ≥ 19 mm (estimated one-year survival 74% vs. 96%, *p* < 0.01) and for patients with moderate/severe TR compared to those with no/trace/mild TR (estimated one-year survival 70% vs. 93%, *p* < 0.01). Furthermore, among intermediate-risk patients, those with both TAPSE < 19 mm and moderate/severe TR had an estimated one-year survival (56%) similar to that of high-risk patients (56%), and those with both TAPSE ≥ 19 mm and no/trace/mild TR had an estimated one-year survival (97%) similar to that of low-risk patients (95%). Conclusions: Echocardiography, a routinely performed, non-invasive imaging modality, plays a pivotal role in discriminating distinct survival phenotypes among prevalent intermediate-risk PAH patients using TAPSE and degree of TR. This can potentially help guide subsequent therapy.

## 1. Introduction

Pulmonary arterial hypertension (PAH) is a rare disease characterized by the chronic remodeling of distal pulmonary arteries, leading to a progressive increase in pulmonary arterial load [1]. Despite recent advances in medical therapy and the development of novel strategies for combination therapy, PAH remains a severe clinical condition with a poor prognosis [2]. During the 6th World Symposium on Pulmonary Hypertension (WSPH), the risk stratification of PAH patients was greatly emphasized, not only to define long-term prognosis but also to guide therapeutic management [3]. Several multiparametric risk stratification approaches have been used to stratify PAH patients into low-, intermediate-, and high-risk based on expected transplant-free survival, with the aim of titrating medical therapy to achieve low-risk status [4,5,6,7,8,9]. However, studies have shown that the majority of PAH patients remained in the intermediate-risk category after receiving initial PAH-specific therapy [5,6,7], including those receiving recommended upfront combination therapy with a phosphodiesterase type 5 inhibitor (PDE5-I) and an endothelin receptor antagonist (ERA) [10,11]. Several treatment options are available for patients who remain in the intermediate-risk category at follow-up [4], and the best treatment strategy for these patients remains uncertain.

The role of 2D echocardiography, a routinely used imaging modality in PAH patients, has been underscored in risk assessment and stratification [3,12]. While both cardiac magnetic resonance and transthoracic echocardiography are non-invasive modalities instrumental for the evaluation of right ventricular (RV) morphology and function with relevant prognostic implications in PAH [13,14,15,16,17], the role of imaging in risk stratification approaches has been limited to the assessment of right atrial (RA) area and the presence of pericardial effusion [4,7].

Given the prognostic value of RV echocardiographic parameters in predicting mortality risk in PAH and the need for a better risk stratification strategy, we hypothesized that echocardiography could be used to further stratify prevalent patients conventionally classified as intermediate-risk into intermediate-low- and intermediate-high-risk groups and thereby potentially help inform treatment decisions. Therefore, the aims of this study were to: (1) compare echocardiographic parameters between PAH risk groups, (2) evaluate the prognostic value of these parameters in predicting mortality, and (3) incorporate identified parameters in the risk stratification of PAH patients. 

## 2. Materials and Methods

### 2.1. Study Population

This is a retrospective analysis of PAH patients previously enrolled in observational studies [13,15,17,18] at Johns Hopkins Pulmonary Hypertension (PH) Center, the University of Pennsylvania Hospital, and Temple University Hospital. Informed consent was obtained from each patient in these studies. PAH was diagnosed by PH specialists based on right heart catheterization (RHC) findings, defined as resting mean pulmonary arterial pressure ≥ 25 mmHg, pulmonary vascular resistance > 3 Wood units, and pulmonary capillary wedge pressure ≤ 15 mmHg [19]. Only patients with prevalent PAH who were on vasodilator therapy were included. Patients < 18 years old at diagnosis, with incident PAH, with WSPH Group 2, 3, 4, or 5 PH [19], or with missing RHC or echocardiographic data were excluded.

### 2.2. Variables of Interest

Follow-up data on clinical and hemodynamic parameters obtained following treatment initiation was pooled from databases of previous studies. These were used for the risk stratification of patients and included: World Health Organization functional class (WHO FC), six-minute walking distance (6MWD), serum brain natriuretic peptide (BNP) or N-terminal prohormone BNP (NT-proBNP), and RHC-derived right atrial pressure (RAP) and cardiac index (CI) [6]. Echocardiographic data was obtained from medical records and included the following parameters: RA, RV end-systolic and end-diastolic areas, RA and RV dilation, RV fractional area change (FAC), tricuspid annular plane systolic excursion (TAPSE), estimated pulmonary arterial systolic pressure (PASP), left ventricular (LV) end-systolic and end-diastolic eccentricity indices, and degree of tricuspid regurgitation (TR). Notably, all those parameters were obtained within one month of follow-up RHC, and for the vast majority of patients, echocardiography and RHC were performed on the same day. 

### 2.3. Outcome of Interest

The primary outcome was transplant-free survival, defined from the date of follow-up echocardiography. Patients were followed for up to 5 years. Data on all-cause mortality was extracted from databases of previous studies.

### 2.4. Statistical Analysis

All statistical analyses were conducted using Stata version 17.0 (StataCorp. College Station, TX, USA). For descriptive analyses, we used mean ± SD and one-way analysis of variance for normally distributed continuous variables and *n* (%) and the chi-square or Fisher exact tests for categorical variables. Univariate Cox proportional hazards models were used to assess the association between echocardiographic variables and survival. Time-to-event analysis was performed using the Kaplan–Meier product-limit estimator to compare survival between patients’ risk strata. The proportional hazards assumption was examined and met for all covariates using a continuous time-varying predictor and generalized linear regression of scaled Schoenfeld residuals on functions of time. The goodness of fit of Cox proportional hazards models for predicting five-year survival was assessed using Harrell’s C-statistic and Akaike information criteria (AIC) [20,21]. The dataset was randomly split into training and test sets, and the discriminatory power of risk stratification models was compared using Harrell’s C and Somers’ D [21]. All comparisons were two-sided, and a *p*-value less than 0.05 was used to represent statistical significance. 

### 2.5. Risk Stratification

Patients were stratified into three risk categories using a validated PAH risk stratification approach devised by the Comparative, Prospective Registry of Newly Initiated Therapies for Pulmonary Hypertension (COMPERA) investigators [6]. A rounded average was computed for every patient based on five clinical and hemodynamic parameters (cutoffs shown in Appendix B
Table A1) and was used to assign risk: 1 for low risk, 2 for intermediate risk, and 3 for high risk. Echocardiographic parameters significantly associated with survival in univariate Cox proportional hazards models were used to further stratify patients in the intermediate-risk group into intermediate-low and intermediate-high groups.

## 3. Results

### 3.1. Study Population and Characteristics

A total of 146 patients (2004–2015) met the eligibility criteria of our study (Appendix A). The demographic characteristics of the cohort are shown in Table 1. The cohort was predominantly white (78%) and female (83%), with an average age of 56 ± 14 years at diagnostic RHC. Most patients (>90%) had either idiopathic PAH or connective-tissue disease-associated PAH (CTD-PAH), with median disease duration of 13 months [IQR, 11–23 months]. All patients were on vasodilator therapy. Eight patients (5.5%) were exclusively on calcium channel blockers; the rest were receiving PAH-specific therapy. Nearly 50% of those receiving PAH-specific therapy were on dual combination therapy.

Clinical and hemodynamic characteristics of the cohort used in risk stratification are shown in Table 2. Nearly half of the cohort had WHO FC I or II symptoms, and one-third had WHO FC III symptoms. Most patients (65%) had a 6MWD between 165 m and 440 m. Around two-thirds of the cohort had serum BNP levels ≤ 300 ng/L or NT-proBNP levels ≤ 1400 ng/L. Most patients had a RAP ≤ 14 mmHg (88%) and a CI ≥ 2 L/min/m^2^ (72%). Additional hemodynamic characteristics of the cohort are shown in Appendix B
Table A2. 

Nearly all patients (97%) had reported data on at least 3 out of the 5 parameters used for risk stratification (Appendix B
Figure A1). Using the COMPERA rounded average approach, 43 (29%) patients were classified as low-risk, 89 (61%) patients as intermediate-risk, and 14 (10%) patients as high-risk.

### 3.2. Outcomes

A total of 38 (30%) patients died over a median follow-up of 2.5 years [IQR, 1.5–4.9 years]. None of the patients underwent lung transplantation during the follow-up period. Overall one-year and three-year transplant-free survival rates were 86% and 74%, respectively. The Kaplan-Meier estimated one-year survival rate was 95% for the low-risk group, 86% for the intermediate-risk group, and 56% for the high-risk group (Figure 1).

### 3.3. Echocardiography, Risk Groups, and Survival

Echocardiographic parameters were compared between risk groups (Table 3). RA, RV end-systolic, and RV end-diastolic areas increased steadily across risk groups (*p* = 0.01, *p* = 0.01, *p* = 0.04, respectively). This was consistent with a higher proportion of patients with reported qualitative moderate/severe RA and RV dilation in the intermediate- and high-risk groups although the difference was not statistically significant (*p* = 0.35, *p* = 0.59, respectively). RV function decreased steadily across risk groups as evidenced by a progressive drop in RV FAC and TAPSE (*p* = 0.01, *p* < 0.01, respectively) [22]. PASP was not different between risk groups (*p* = 0.26). Although not statistically significant, there was a tendency towards more RV pressure and volume overload going from low-risk to high-risk, as reflected by a progressive increase in LV end-systolic and end-diastolic eccentricity indices (*p* = 0.06, *p* = 0.16, respectively). Finally, patients in the intermediate-/high-risk groups were more likely to have moderate/severe TR, as compared to those in the low-risk groups (*p* = 0.04).

The association between echocardiographic parameters and survival was assessed using univariate Cox proportional hazards models (Table 4). Although patients with larger RA and RV areas, moderate/severe RA and RV dilation, lower RV fractional area change, and higher PASP and LV eccentricity indices tended to have worse survival, only TAPSE (HR 1.24 per 2 mm decline, 95% CI 1.08–1.43, *p* < 0.01) and degree of TR (HR 3.27 for moderate/severe TR, 95% CI 1.72–6.23, *p* < 0.01) were significantly associated with survival among the measured echocardiographic parameters.

### 3.4. Use of Echocardiography for Risk Stratification

Echocardiographic parameters significantly associated with survival (TAPSE and degree of TR) were used to further stratify patients in the intermediate-risk group into intermediate-low- and intermediate-high-risk groups. To determine the optimal TAPSE cutoff for risk stratification, patients were stratified using different TAPSE cutoffs falling into the TAPSE IQR [16–22 mm] and spaced at 1 mm intervals. Survival analyses were conducted to compare survival for patients with TAPSE at/above the cutoff vs. those with TAPSE below the cutoff, and the *p*-value for the log-rank test was reported for every cutoff. A TAPSE value of 19 mm provided the highest predictive value, i.e., the lowest *p*-value for the log-rank test (Appendix B
Figure A2) and coincided with the median, and thus was chosen as the stratification cutoff.

Using TAPSE, intermediate-risk patients were stratified into intermediate-low-risk (TAPSE ≥ 19 mm) and intermediate-high-risk (TAPSE < 19 mm) groups. Patients in the intermediate-low-risk group had significantly better survival compared to those in the intermediate-high-risk group (estimated one-year survival 96% vs. 74%, *p* < 0.01; Figure 2). Furthermore, this four-strata approach had a significantly higher discriminatory power compared to the conventional three-strata approach (Harrell’s C 0.80, AIC 330 vs. Harrell’s C 0.72, AIC 340, *p* < 0.01).

Using a similar approach, the degree of TR was used to stratify intermediate-risk patients into intermediate-low-risk (no/trace/mild TR) and intermediate-high-risk (moderate/severe TR) groups. Patients in the intermediate-low-risk group had significantly better survival compared to those in the intermediate-high-risk group (estimated one-year survival 93% vs. 70%, *p* < 0.01; Figure 3). Although this four-strata approach tended to have higher discriminatory power compared to the conventional three-strata approach, the difference was not statistically significant (Harrell’s C 0.76, AIC 334 vs. Harrell’s C 0.71, AIC 340, *p* = 0.12).

Finally, the combination of both TAPSE and degree of TR was used for risk stratification. Among intermediate-risk patients, survival was best for patients with TAPSE ≥ 19 mm and no/trace/mild TR, worse for patients with either TAPSE < 19 mm or moderate/severe TR, and worst for patients with TAPSE < 19 mm and moderate/severe TR (estimated one-year survival 97% vs. 87% vs. 56%, respectively, *p* < 0.01; Figure 4). This five-strata model also had significantly higher discriminatory power compared to the conventional three-strata model (Harrell’s C 0.81, AIC 326 vs. Harrell’s C 0.72, AIC 340, *p* < 0.01).

## 4. Discussion

In this study, echocardiography was used to refine an existing PAH risk stratification model and outline a more nuanced approach for the risk assessment of intermediate-risk patients. Our main findings were: (1) several echocardiographic parameters related to RV structure and function (RA area, RV end-systolic and end-diastolic areas, RV FAC, TAPSE, degree of TR) correlated strongly with PAH risk strata, with worse parameters seen in intermediate- and high-risk patients; (2) among a panel of echocardiographic parameters, TAPSE and degree of TR were significantly and strongly associated with transplant-free survival; and (3) echocardiography improves the risk stratification of PAH patients, as it allows for the further stratification of intermediate-risk patients using TAPSE and/or degree of TR. This is the first study to our knowledge that demonstrates an added value from conventional echocardiographic measurements to existing multiparametric risk stratification of patients with prevalent PAH receiving stable PAH therapy.

Echocardiography is a non-invasive, widely accessible clinical tool that is suggested as the first diagnostic study when PAH, or generally PH, is suspected [4,5,22]. Once PAH is diagnosed, echocardiography is recommended every 6–12 months and even earlier after changes in therapy or clinical worsening [4,5]. Despite this, the role of echocardiography in the risk stratification of patients with PAH has often been underscored and limited to two echocardiographic parameters in conventional risk stratification approaches: RA area [4,7] and the presence of pericardial effusion [4,7,8]. Such parameters are indirect manifestations of cardiac dysfunction, and their prognostic value has not been consistent across studies [23]. In patients with PAH, symptoms and prognosis are largely reflected by RV morphology and function, and its adaptation to increased afterload [24,25]. From a pathophysiologic standpoint, elevated afterload has a progressive negative impact on right heart function, ultimately leading to right heart failure from ventriculoarterial uncoupling [26]. Several studies have demonstrated a strong association between outcomes in PAH cohorts and echocardiographic measurements that reflect RV function, including TAPSE [13,14], TAPSE/PASP [27,28], LV end-diastolic eccentricity index [14], degree of TR [14], global RV longitudinal systolic strain and strain rate [15,29], and RV post-systolic strain patterns derived from the mid-basal RV free wall segments [16]. We confirmed many of these associations (TAPSE, TAPSE/PASP, degree of TR) in our study and showed that their clinical utility can be extrapolated to refine the COMPERA risk stratification approach, which combines clinical, functional, exercise, and hemodynamic parameters to stratify patients into low-, intermediate-, and high-risk for one-year mortality [6]. In this study, we describe a refined approach to further stratify intermediate-risk patients using echocardiographic parameters.

Risk stratification of patients with PAH is essential for guiding therapy, with the goal of achieving or maintaining a low-risk profile [3,4]. However, several studies have shown that most patients, including those treated with dual oral combination therapy, remain in the intermediate-risk category, although it is unlikely that they all share the same mortality risk [5,6,7,10,11]. This is important from a clinical standpoint since the management of patients with intermediate risk is not straightforward, unlike that of patients with low-risk who need no further change in therapy or those with high-risk who obviously require more aggressive interventions, such as parenteral therapy or lung transplantation referral [4,30]. Therapeutic options for patients who remain in the intermediate-risk category despite dual oral combination therapy with a PDE5-I and an ERA include adding selexipag [31,32], switching from PDE5-I to riociguat [33,34,35], initiating parenteral prostanoids [12] and lung transplantation referral [12]. Based on this, we believe that the further risk stratification of patients in the intermediate-risk category may help clinicians decide how aggressive subsequent interventions should be. The current study indicates that TAPSE and degree of TR can differentiate patients with preserved cardiac function and better prognosis from those with poor cardiac function and worse prognosis within the intermediate-risk group. Notably, patients in the intermediate-risk group who had both TAPSE ≥ 19 mm and no/trace/mild TR had similar estimated one-year survival (97%) as those in the low-risk group (95%) and thus probably require minimal therapeutic intervention. Likewise, patients in the intermediate-risk group who had both TAPSE < 19 mm and moderate/severe TR had similar estimated one-year survival (56%) as those in the high-risk group (56%) and would require more aggressive intervention.

With the majority of patients remaining in the intermediate-risk category after initial therapy, more recent studies modified existing approaches to provide a more granular risk stratification of patients with PAH. Recent data suggested that stroke volume index at first follow-up following the diagnosis of PAH can discriminate between two survival phenotypes within the intermediate-risk category [36]. Additionally, a four-strata approach described by COMPERA investigators computes rounded average risk scores using refined cut-off levels for WHO FC, 6MWD, and BNP/NT-proBNP and provides better discrimination compared to the conventional three-strata approach using the same variables [37,38]. Another recent study identified a subgroup of intermediate-risk patients with 6MWD ≥ 270 m and TAPSE/PASP ≥ 0.24 mm/mmHg who had significantly better survival compared to other patients in the same risk group [28]. While these studies allowed for more granular risk stratification, they relied heavily on WHO FC, a very subjective assessment [39], and on 6MWD, which has questionable validity in CTD-PAH due to comorbid musculoskeletal limitations [40]. The current study presents a new four-strata approach that allows for better discrimination by incorporating more objectively obtained clinical echocardiographic parameters into the existing COMPERA model. While both TAPSE and degree of TR were used to further stratify intermediate-risk patients, the approach that combined both parameters had the highest discriminatory power, with a C-statistic (0.81) exceeding that of conventional three-strata approaches (COMPERA score C-statistic = 0.62, French Pulmonary Hypertension Registry model C-statistic = 0.64, Registry to Evaluate Early and Long-Term PAH Disease Management (REVEAL) 2.0 score C-statistic = 0.73) [5,6,41], and, more importantly, that of novel four-strata risk scores (COMPERA 2.0 score C-statistic = 0.73, Yogeswaran et al. model C-statistic = 0.72) [28,37,38]. A recent study of 102 PAH patients showed that TAPSE/TR velocity and TAPSE/PASP have the potential for dichotomizing intermediate-risk patients into two risk categories [42]. However, the majority of patients (75%) in this study had incident PAH and were not on PAH therapy, unlike patients included in our study who were already on optimal PAH therapy. This is clinically important, as the management of intermediate-risk patients with incident PAH may be straightforward, unlike that of prevalent PAH patients, who remain intermediate-risk despite receiving PAH therapy.

Our study had a number of limitations. Our cohort was enriched with connective tissue disease-associated PAH, which may limit the generalizability of the findings. Although patients were prospectively enrolled in the registry, the analysis was retrospective in nature and was limited to a relatively small sample size, which limited additional subgroup analyses. Although the COMPERA model includes mixed venous oxygen saturation (MvO2) as one of the parameters used for risk stratification, it was not available for patients in our cohort and hence not included. However, we believe this has minimal effect on our results as the COMPERA model computes an averaged risk for every patient provided that at least two parameters are available, a condition which was satisfied by all patients in our cohort, with the vast majority having at least three parameters. The echocardiographic parameters used in risk stratification are clinically available, yet they are prone to inherent inter-reader variability, especially the degree of TR, which may be under- or overestimated [43]. A further limitation is that echocardiographic data was collected in different centers with no core measurement and over a 15-year period, with some being obtained before the publication of the Guidelines for Echocardiographic Assessment of the Right Heart in Adults, which standardized right heart echocardiographic evaluation [22]. Nevertheless, all patients were evaluated in PAH referral centers by expert echocardiographers. Finally, given our sample size and the limited number of subjects reaching the primary outcome during the follow-up period, we could not run extensive multivariate Cox regression analysis and adjust for all potential confounders.

## 5. Conclusions

In summary, echocardiography is a powerful, non-invasive, routinely performed prognostic tool that has clinical utility in the additional risk stratification of patients with PAH at intermediate-risk. In this study, we identified TAPSE and degree of TR as two independent predictors of mortality in prevalent PAH and used these two simple echocardiographic parameters to further stratify intermediate-risk patients according to five-year mortality risk. We believe this can potentially help guide therapeutic decisions in these patients.

## Figures and Tables

**Figure 1 jcm-11-04034-f001:**
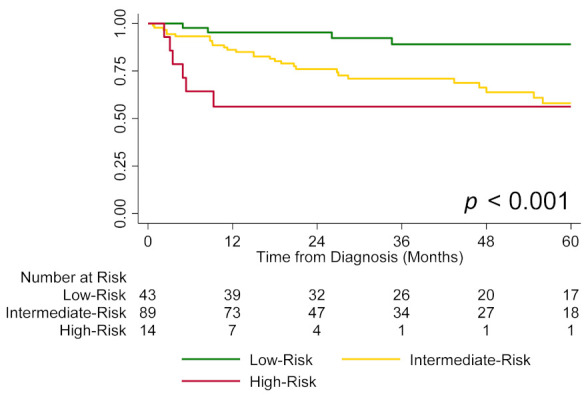
Transplant-free survival by risk group (low vs. intermediate vs. high).

**Figure 2 jcm-11-04034-f002:**
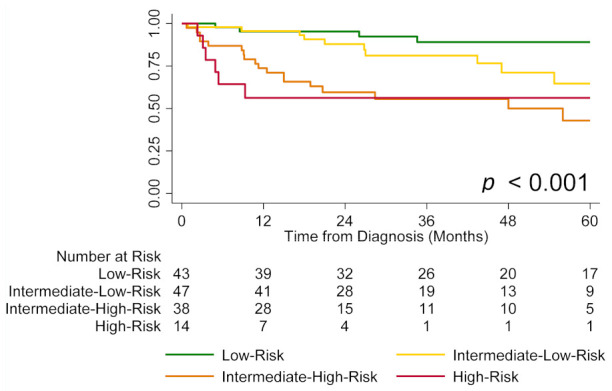
Transplant-free survival by risk group (low vs. intermediate-low vs. intermediate-high vs. high) using tricuspid annular plane systolic excursion (TAPSE) to further stratify intermediate-risk patients. Note: Four intermediate-risk patients had missing TAPSE and were not included in the analysis.

**Figure 3 jcm-11-04034-f003:**
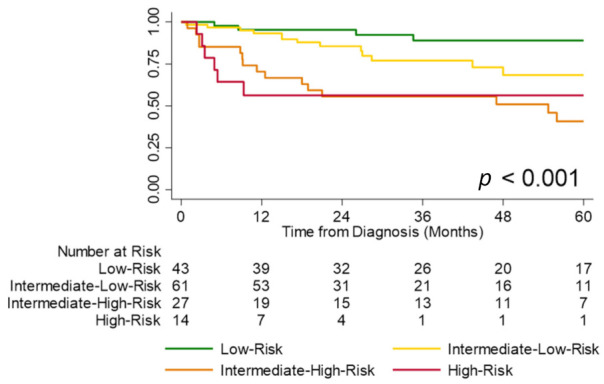
Transplant-free survival by risk group (low vs. intermediate-low vs. intermediate-high vs. high) using degree of tricuspid regurgitation (TR) to further stratify intermediate-risk patients. Note: One intermediate-risk patient missed their degree of TR and was not included in the analysis.

**Figure 4 jcm-11-04034-f004:**
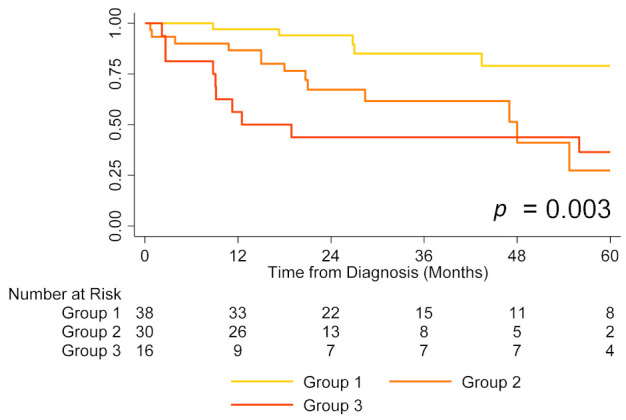
Transplant-free survival by risk group using both tricuspid annular plane systolic excursion (TAPSE) and degree of tricuspid regurgitation (TR) to further stratify intermediate-risk patients (Group 1: TAPSE ≥ 19 mm and no/trace/mild TR; Group 2: TAPSE < 19 mm or moderate/severe TR; Group 3: TAPSE < 19 mm and moderate/severe TR). Note: Graphs for low- and high-risk patients are not shown to avoid crowding the figure; five intermediate-risk patients had missing TAPSE or degree of TR and were not included in the analysis.

**Table 1 jcm-11-04034-t001:** Patients Characteristics.

		Mean ± SD or *n* (%); *n*= 146
**PAH Diagnosis Age, Years**		56 ± 14
**Female**		121 (82.9)
**Race**	White	114 (78.1)
	African American	24 (16.4)
	Other	8 (5.5)
**PAH Subtype**	IPAH	68 (46.6)
	CTD-PAH	67 (45.9)
	Other	11 (7.5)
**Treatment**	CCB only	8 (5.5)
	PAH-specific therapy	138 (94.5)
	Single/dual/triple *	51 (36.9)/65 (47.1)/22 (16)
	PDE5-I/ERA/prostanoids *	109 (79)/91 (65.9)/47 (34.1)

* Only among patients receiving PAH-specific therapy (*n* = 138). Abbreviations: PAH = pulmonary arterial hypertension; IPAH = idiopathic pulmonary arterial hypertension; CTD-PAH = connective tissue disease-associated pulmonary arterial hypertension; CCB = calcium channel blockers; PDE5-I = phosphodiesterase type 5 inhibitor; ERA = endothelin receptor antagonist.

**Table 2 jcm-11-04034-t002:** Clinical and Hemodynamic Characteristics of the Cohort Used in Risk Stratification.

		*n* (%)
**WHO FC [*n* = 146]**	I/II	72 (49.3)
	III	49 (33.6)
	IV	25 (17.1)
**6-MWD, m [*n* = 133]**	>440 m	36 (27.1)
	165–440 m	86 (64.7)
	<165 m	11 (8.3)
**NT-proBNP/BNP, ng/L [*n* = 101]**	<300/<50	28 (27.7)
	300–1400/50–300	41 (40.6)
	>1400/>300	32 (31.7)
**RAP, mmHg [*n* = 109]**	<8	50 (45.9)
	8–14	46 (42.2)
	>14	13 (11.9)
**CI, L/min/m^2^ [*n* = 105]**	≥2.5	54 (51.4)
	2–2.4	27 (25.7)
	<2	24 (22.9)

Abbreviations: WHO FC = World Health Organization Functional Class; 6-MWD = 6-minute walking distance; NT-proBNP = N-terminus pro-brain natriuretic peptide; BNP = brain natriuretic peptide; RAP = right atrial pressure; CI = cardiac index.

**Table 3 jcm-11-04034-t003:** Comparison of Echocardiographic Parameters between Risk Groups.

	Mean ± SD or *n* (%)			
	Low	Intermediate	High	*p*-Value †
**RA Area, cm^2^**	18.9 ± 6.2 [*n* = 36]	21.3 ± 7.6 [*n* = 67]	27.4 ± 7.4 [*n* = 8]	0.011
**RV End-Systolic Area, cm^2^**	18.9 ± 7.1 [*n* = 35]	21.1 ± 8.4 [*n* = 63]	29.0 ± 7.4 [*n* = 7]	0.010
**RV End-Diastolic Area, cm^2^**	26.5 ± 8.0 [*n* = 35]	28.5 ± 9.1 [*n* = 63]	35.7 ± 6.9 [*n* = 7]	0.041
**Moderate/Severe RA Dilation * (vs. None/Mild)**	12 (33.3) [*n* = 36]	26 (38.8) [*n* = 67]	5 (62.5) [*n* = 8]	0.353
**Moderate/Severe RV Dilation ° (vs. None/Mild)**	9 (25.7) [*n* = 35]	20 (31.8) [*n* = 63]	3 (42.9) [*n* = 7]	0.587
**RV FAC, %**	30.1 ± 11.2[*n* = 42]	27.8 ± 11.4 [*n* = 87]	19.6 ± 9.9 [*n* = 14]	0.011
**TAPSE, mm**	19.6 ± 4.0 [*n* = 39]	19.2 ± 5.1[*n* = 85]	13.4 ± 3.1 [*n* = 14]	<0.001
**PASP, mmHg**	47.5 ± 24.2 [*n* = 30]	56.2 ± 24.9 [*n* = 60]	52.8 ± 8.8 [*n* = 10]	0.260
**TAPSE/PASP, mm/mmHg**	0.65 ± 0.53 [*n* = 27]	0.42 ± 0.28 [*n* = 56]	0.26 ± 0.05 [*n* = 10]	0.005
**LVED Eccentricity Index**	1.04 ± 0.30 [*n* = 22]	1.18 ± 0.35 [*n* = 54]	1.27 ± 0.43 [*n* = 9]	0.156
**LVES Eccentricity Index**	0.98 ± 0.31 [*n* = 24]	1.36 ± 0.87 [*n* = 46]	1.88 ± 2.05 [*n* = 8]	0.056
**Moderate/Severe TR ” (vs. None/Trace/Mild)**	11 (25.6) [*n* = 43]	27 (30.7) [*n* = 88]	8 (61.5) [*n* = 13]	0.044

* Defined as RA area > 22 cm^2^ measured from apical 4-chamber view. ° Defined as RV end-diastolic area >33 cm^2^ measured from apical right ventricle-focused 4-chamber view. ” Measured using the color flow doppler 2D modality (jet area, vena contracta) and continuous wave doppler (density of regurgitant jet). † For the one-way analysis of variance (ANOVA) test. Abbreviations: RA = right atrial; RV = right ventricular; FAC = fractional area change; TAPSE = tricuspid annular plane systolic excursion; PASP = pulmonary arterial systolic pressure; LVED = left ventricular end-diastolic; LVES = left ventricular end-systolic; TR = tricuspid regurgitation.

**Table 4 jcm-11-04034-t004:** Associations between Echocardiographic Parameters and Survival.

	HR (95% CI) *	*p*-Value
**RA area, per 1 cm^2^ increase**	1.03 (0.98–1.07)	0.239
**RV end-systolic area, per 1 cm^2^ increase**	1.04 (0.99–1.08)	0.119
**RV end-diastolic area, per 1 cm^2^ increase**	1.04 (1.00–1.08)	0.065
**Moderate/severe RA dilation (vs. none/mild)**	1.41 (0.64–3.12)	0.393
**Moderate/severe RV dilation (vs. none/mild)**	1.27 (0.55–2.95)	0.579
**RV FAC, per 5% decrease**	1.10 (0.96–1.28)	0.177
**TAPSE, per 2 mm decrease**	1.24 (1.08–1.43)	0.002
**PASP, per 5 mmHg**	1.04 (0.96–1.12)	0.349
**TAPSE/PASP, per 1 mm/5 mmHg decrease**	1.5 (1.04–2.11)	0.03
**LVED Eccentricity Index, per unit increase**	2.5 (0.84–7.36)	0.100
**LVES Eccentricity Index, per unit increase**	1.21 (0.86–1.72)	0.279
**Moderate/severe TR (vs. none/trace/mild)**	3.27 (1.72–6.23)	<0.001

* Adjusting for age, sex, and PAH etiology (connective tissue disease-associated vs. other), all assessed echocardiographic parameters were not significantly associated with transplant-free survival, except: TAPSE (HR 1.16 per 2 mm decrease, 95% CI 1.00–1.35, *p* = 0.046), moderate/severe TR (HR 3.00, CI 1.53–5.92, *p* = 0.001, and RV end-diastolic area (HR 1.05, CI 1.00–1.10, *p* = 0.042). Abbreviations: RA = right atrial; RV = right ventricular; FAC = fractional area change; TAPSE = tricuspid annular plane systolic excursion; PASP = pulmonary arterial systolic pressure; LVED = left ventricular end-diastolic; LVES = left ventricular end-systolic; TR = tricuspid regurgitation.

## Data Availability

The data presented in this study are available on request from the corresponding author.

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
