# Peer review of "Risk Stratification of Patients with Pulmonary Arterial Hypertension: The Role of Echocardiography"

_jcm, 2022, doi:10.3390/jcm11144034_

Round 1

Reviewer 1 Report

In this manuscript, Mercurio et al. explored the utility of echocardiographic parameters in providing incremental risk assessment to established risk scores for patients with pulmonary arterial hypertension (PAH). 146 patients with prevalent PAH were identified from ongoing observational studies and risk stratified into low-, intermediate-, and high risk categories using the well-validated COMPERA risk score. In a univariate analysis, TAPSE and degree of tricuspid regurgitation were the strongest predictors of survival and were subsequently used to further risk stratify 89 patients in the intermediate risk category to intermediate low and intermediate high risk categories. Patients with both TAPSE < 19 mm and more severe tricuspid regurgitation had the worse outcomes. The authors concluded that the addition of these echocardiographic variables to established risk scores such as COMPERA can aid in therapeutic approach.

This is a well written manuscript with a clear clinical question and appropriate methodology. This study would add greatly to the current literature given that many echocardiographic variables are shown to be strong prognostic markers in patients with PAH and should be used in the initial risk assessment. There are only a few comments/suggestions:

1.     When were the echocardiograms performed in relation to the right heart catheterizations in which the data was obtained for risk stratification?

2.     How many PAH patients were without echocardiographic data from these observational studies and what was the reason for not having this data. In other words, did poor echocardiographic images play a role?

3.     How was right atrial and right ventricle dilation measured?

4.     How was the degree of tricuspid regurgitation measured?

5.     In the limitations, would recommend that the authors underscore the point that echo parameters were acquired at different sites by different sonographers and from different decades (before time of right heart guidelines) which really increases variability in data. The authors note this but it should be highlighted as it might directly affect their results

6.     When exactly were patients stratified into three risk categories using COMPERA? Which right heart catheterization was used for the hemodynamics? At what time point was NT-proBNP/BNP measured? When was functional capacity assessed? Was is after they were already on vasodilators? Before drugs were initiated?

7.     The total sample size (n=146) seems low especially since patients were selected across four separate observational studies from three separate sites. Can the authors comment on this?

8.     For table 3, the p-value is for what specifically?

9.     For table 3, it might be nice to show something similar for after the population is divided into 4 risk categories

10.  From table 4, it appears that TAPSE/PASP was also significantly associated with survival. Why was this not included in the risk stratification?

11.  From table 4, the HR for TAPSE, per 2 mm decrease is 1.24 but the text (page 4, line 178) suggests it is 1.08. Can the authors explain the discrepancy?

12.  It might be interesting to see how these echocardiography parameters modify risk using other validated risk stratification measures such as REVEAL, the French PH Network, and the Swedish PAH Registry. Why did the authors choose to only use COMPERA?

Author Response

Reviewer 1

We thank the reviewer for taking the time to review the manuscript and share his/her helpful feedback.

Comments and Suggestions for Authors:

In this manuscript, Mercurio et al. explored the utility of echocardiographic parameters in providing incremental risk assessment to established risk scores for patients with pulmonary arterial hypertension (PAH). 146 patients with prevalent PAH were identified from ongoing observational studies and risk stratified into low-, intermediate-, and high risk categories using the well-validated COMPERA risk score. In a univariate analysis, TAPSE and degree of tricuspid regurgitation were the strongest predictors of survival and were subsequently used to further risk stratify 89 patients in the intermediate risk category to intermediate low and intermediate high risk categories. Patients with both TAPSE < 19 mm and more severe tricuspid regurgitation had the worse outcomes. The authors concluded that the addition of these echocardiographic variables to established risk scores such as COMPERA can aid in therapeutic approach.

This is a well written manuscript with a clear clinical question and appropriate methodology. This study would add greatly to the current literature given that many echocardiographic variables are shown to be strong prognostic markers in patients with PAH and should be used in the initial risk assessment. There are only a few comments/suggestions:

  1. When were the echocardiograms performed in relation to the right heart catheterizations in which the data was obtained for risk stratification?

Response: Thank you for bringing up this important point. Echocardiography and RHC were performed on the same day for the vast majority of patients and within one month for all the patients. We believe this should be mentioned in the manuscript, and we did so in the second paragraph of the Methods section.

  1. How many PAH patients were without echocardiographic data from these observational studies and what was the reason for not having this data. In other words, did poor echocardiographic images play a role?

Response: We thank the reviewer for pointing this out. We know that two patients were excluded from the analysis in the Forfia et al. paper for technically inadequate echocardiographic windows. We do not have such information from the other studies, but we think that would be the case for a few other parameters. Nevertheless, we think that this likely had minimal impact on our results as reflected by the low extent of missingness in Table 3 (e.g., 8 patients did not have TAPSE, 2 patients did not have degree of tricuspid regurgitation).

  1. How was right atrial and right ventricle dilation measured?

Response: Moderate/Severe right atrial dilatation was defined as a right atrial area >22 cm2 measured from apical 4-chamber view. Moderate/Severe right ventricular dilation was defined as right ventricular end-diastolic area > 33 cm2 obtained on apical right ventricle-focused 4-chamber view. We think that adding this to our paper would make it clearer, and we did so in the footnote of Table 3. We thank the reviewer for touching on this.

  1. How was the degree of tricuspid regurgitation measured?

Response: This is absolutely an important point that should be mentioned in our paper. The degree of tricuspid regurgitation was measured by means of a qualitative assessment distinguishing no/trace/mild tricuspid regurgitation vs. moderate/severe tricuspid regurgitation using the color flow doppler 2D modality (jet area, vena contracta) and continuous wave doppler (density of regurgitant jet). We added this to the footnote of Table 3.

  1. In the limitations, would recommend that the authors underscore the point that echo parameters were acquired at different sites by different sonographers and from different decades (before time of right heart guidelines) which really increases variability in data. The authors note this, but it should be highlighted as it might directly affect their results

Response: We thank the reviewer for this observation. As suggested, we added the following sentence to the Limitations section: “A further limitation is that echocardiographic data was collected in different centers with no core measurement and over a 15-year period, with some being obtained before the publication of the Guidelines for Echocardiographic Assessment of the Right Heart in Adults, which standardized right heart echocardiographic evaluation. Nevertheless, all patients were evaluated in PAH referral centers by expert echocardiographers.”

  1. When exactly were patients stratified into three risk categories using COMPERA? Which right heart catheterization was used for the hemodynamics? At what time point was NT-proBNP/BNP measured? When was functional capacity assessed? Was is after they were already on vasodilators? Before drugs were initiated?

Response: We thank the reviewer for this comment. All patients were stratified into risk categories at follow-up after receiving PAH therapy. All parameters used in risk stratification were assessed at follow-up, and were evaluated within a month of each other, including RHC, echocardiography, BNP/NT-proBNP, and WHO FC. All patients were stable on vasodilators at the time of risk assessment. We definitely agree that we should make this clearer in our paper. We added these details to the second paragraph of the Methods section.

  1. The total sample size (n=146) seems low especially since patients were selected across four separate observational studies from three separate sites. Can the authors comment on this?

Response: This is definitely a crucial observation. For each of the 4 observational studies, we detailed the number of patients included in the original study and the number of those excluded from the current study, with the reasons for exclusion (Supplemental Table S1 ). The main reasons for exclusion were overlapping subjects and incident patients.

  1. For table 3, the p-value is for what specifically?

Response: The p-value in Table 3 is for the one-way analysis of variance (ANOVA) test comparing echocardiographic parameters between the 3 risk groups. We added this to the footnote of Table 3 to make it clearer.

  1. For table 3, it might be nice to show something similar for after the population is divided into 4 risk categories

Response: We appreciate the reviewer’s feedback. We definitely agree that it would be interesting to add a similar table showing the differences in echocardiographic parameters between the 4 risk groups. A major barrier to conducting this type of analysis is that we used 3 methods for further risk stratification: TAPSE, degree of TR, and TAPSE + degree of TR combined. Hence, we would have to add 3 tables comparing echocardiographic parameters for each method used. In order not to make the manuscript more crowded and keep our message straightforward, we opted not to add these tables.

  1. From table 4, it appears that TAPSE/PASP was also significantly associated with survival. Why was this not included in the risk stratification?

Response: This is surely an accurate observation. Since TAPSE/PASP is a ratio that is dependent on both TAPSE and PASP, and since TAPSE but not PASP was significantly different between the risk groups, we believe the statistically significant difference observed for the ratio is mostly driven by TAPSE. It would also be redundant to further stratify patients by TAPSE then by TAPSE/PASP. Furthermore, TAPSE and degree of TR both had stronger predictive values (P<0.01) compared to TAPSE/PASP. Lastly, TASPSE/PASP was no longer associated with survival after adjusting for age, sex, and PAH type, unlike TAPSE and degree of TR. For these reasons, TAPSE/PASP was not included in further risk stratification.

  1. From table 4, the HR for TAPSE, per 2 mm decrease is 1.24 but the text (page 4, line 178) suggests it is 1.08. Can the authors explain the discrepancy?

Response: We thank the reviewer for this observation. HR is 1.24. We changed the text appropriately.

  1. It might be interesting to see how these echocardiography parameters modify risk using other validated risk stratification measures such as REVEAL, the French PH Network, and the Swedish PAH Registry. Why did the authors choose to only use COMPERA?

Response: We thank the reviewer for this suggestion. The COMPERA approach uses a “score and average” approach for risk stratification, wherein patients can still be risk stratified even if some of their clinical parameters were missing, unlike other approaches such as the FPHN and REVEAL. For example, if a patient has data on WHO FC, RAP, CI, SvO2, and BNP but missing data on 6MWD, this patient can still be assigned to a risk group using COMPERA, but not using FPHN or REVEAL. Hence, COMPERA would allow us to maintain the power of the study without having to exclude patients because of missing clinical parameters. This is particularly important in our study since data from 4 observational studies was pooled, which would make pooled data very heterogenous. For this reason, we chose to use COMPERA. Nevertheless, we would like to note that we obtained similar results when we used a modified FPHN approach (https://doi.org/10.1164/ajrccm-conference.2020.201.1_MeetingAbstracts.A3846), but we opted not to make our paper crowded and keep the message straightforward by using COMPERA only for risk stratification.

Reviewer 2 Report

In a study by Mercurio et al., the authors aimed to determine whether echocardiographic parameters could improve the risk stratification of intermediate-risk PAH patients. They concluded that echocardiography plays a pivotal role in discriminating distinct survival phenotypes among prevalent intermediate-risk PAH patients using TAPSE and degree of TR. Overall, the manuscript is well-written, and the methodology is well-described.  The study is subject to a few limitations, which have been acknowledged in the manuscript. The authors might consider a few comments to improve the clarity of the manuscript.

- Similar studies have assessed the role of echocardiography, particularly TAPSE and TR severity, in improving risk stratification in PAH patients. The authors might explain the novel aspects of their study compared to previous similar studies and justify the need for their research in current literature. The authors have also claimed that “This is the first study to our knowledge that demonstrates an added value of conventional echocardiographic measurements to existing multiparametric risk stratification of patients with PAH,” but a few previous attempts can be found in the literature (such as DOI: 10.5281/zenodo. 5574447, DOI: 10.1177/2045894020961739, etc.).

- - It is recommended that the authors provide statistical justification for estimating the sample size.

- - The role of potential confounders (such as the type of PAH, age, medical treatment etc.) on the survival of PAH patients has been overlooked in the present study.

The authors might assess the association of TAPSE and degree of TR with survival using the multivariate cox proportional hazards model, adjusting for potential confounders.

-   - The timing of the study is not apparent.

-          - The type of study (i.e., retrospective) should be clearly mentioned in the methods.

Author Response

Reviewer 2

We thank the reviewer for taking the time to review the manuscript and share his/her helpful feedback.

Comments and Suggestions for Authors:

In a study by Mercurio et al., the authors aimed to determine whether echocardiographic parameters could improve the risk stratification of intermediate-risk PAH patients. They concluded that echocardiography plays a pivotal role in discriminating distinct survival phenotypes among prevalent intermediate-risk PAH patients using TAPSE and degree of TR. Overall, the manuscript is well-written, and the methodology is well-described.  The study is subject to a few limitations, which have been acknowledged in the manuscript. The authors might consider a few comments to improve the clarity of the manuscript.

  1. Similar studies have assessed the role of echocardiography, particularly TAPSE and TR severity, in improving risk stratification in PAH patients. The authors might explain the novel aspects of their study compared to previous similar studies and justify the need for their research in current literature. The authors have also claimed that “This is the first study to our knowledge that demonstrates an added value of conventional echocardiographic measurements to existing multiparametric risk stratification of patients with PAH,” but a few previous attempts can be found in the literature (such as DOI: 10.5281/zenodo. 5574447, DOI: 10.1177/2045894020961739, etc.).

Response: We thank the reviewer for the comments and for the time spent reviewing our paper.  In our manuscript, we focused on prevalent PAH patients. Most of prevalent PAH patients reach or remain in the intermediate risk-category (in our cohort 61% of patients were in the intermediate-risk category according to the COMPERA method). In recent years, different therapeutic approaches have been proposed for these patients (adding a third oral drug like Selexipag, switching from PDE5I to Riociguat, adding parenteral prostanoids), but no unique answer has been given. We believe that further stratification of patients in the intermediate-risk category might be useful for clinicians to choose the best course of action for each individual patient. In the present study we performed a risk stratification using the COMPERA method, then added echocardiographic parameters to the COMPERA risk stratification tool to further stratify intermediate-risk patients. This is the first study enrolling exclusively prevalent patients already on PAH treatment. We made this clearer in the first paragraph of the Discussion section. As stated above, one of the biggest challenges for clinicians nowadays is to identify the most appropriate treatment for PAH patients who remain in intermediate-risk despite medical treatment. We already discussed the results of the first study mentioned (Yogeswaran et al, Pulmonary Circulation 2020 - Reference 28). We also cited the other study (Vicenzi et al, Plos One 2022 - Reference 42) and added the following statements to the fourth paragraph of the Discussion section:

“A recent study of 102 PAH patients showed that TAPSE/TR velocity and TAPSE/PASP have the potential of dichotomizing intermediate-risk patients into two risk categories. However, the majority of patients (75%) in this study had incident PAH and were not on any PAH therapy, unlike patients included in our study who were already on optimal PAH therapy. This is clinically important as management of intermediate-risk patients with incident PAH may be straightforward, unlike that of prevalent PAH patients who remain in intermediate-risk despite receiving PAH therapy.”

These two studies are definitely relevant to our study and we thank the reviewer for highlighting them.

  1. It is recommended that the authors provide statistical justification for estimating the sample size.

Response: Thank you for your feedback. Since our study is retrospective in nature, we tried to include all subjects who met our inclusion criteria from the 4 previously published observational studies. We did not do any pre-study sample size estimates. Such estimates would help in determining the sample size needed for the study to be powered enough to detect statistically significant differences if existent. Given the results we obtained, we believe the study was powered enough to address our hypothesis (echocardiography helps improve risk stratification for prevalent PAH patients).   

  1. The role of potential confounders (such as the type of PAH, age, medical treatment etc.) on the survival of PAH patients has been overlooked in the present study. The authors might assess the association of TAPSE and degree of TR with survival using the multivariate cox proportional hazards model, adjusting for potential confounders.

Response: This is absolutely true. We thank the reviewer for highlighting this point. Given our sample size and the number of subjects reaching the primary outcome during the follow-up period (N=38), we could not run extensive multivariate Cox regression analysis and adjust for all potential confounders. The maximum number of covariates that could be included without losing the ability to reliably estimate regression coefficient would be 3-4. Hence, and based on the reviewer’s feedback, we added the results of multivariable regression analysis adjusting for age, sex, type of PAH (CTD-associated vs. other), and one echocardiographic parameter at a time. Age, sex, and type of PAH were chosen as they have been previously shown to be associated with survival and were available for all patients (no missing data). Ideally, other clinical and hemodynamic parameters could be added, although most of those may be associated/collinear with at least one of the echocardiographic parameters being evaluated. Interestingly, the associations observed between TAPSE/degree of TR and survival were still observed even after adjusting for other covariates. We added the results of multivariable analysis to the footnote of Table 4, and we acknowledged the inability to include more covariates in our regression analysis in the limitations section. We thank the reviewer for bringing this to our attention. Indeed, this makes our paper stronger and clearer.

  1. The timing of the study is not apparent.

Response: Thank you for pointing this out. We added the enrollment period (2004-2015) to the first paragraph of the Results section. Additionally, we detailed the enrollment period of each of the 4 observational studies from which we pooled our data in Supplemental Table S1. Regarding the timing of clinical studies relative to each other, we detailed this in the second paragraph of the Methods section as follows:

“Notably, all those parameters were obtained within one month of follow-up RHC, and for the vast majority of patients, echocardiography and RHC were performed on the same day.

  1. The type of study (i.e., retrospective) should be clearly mentioned in the methods.

Response: This should definitely be stated clearly. We added this information to the first paragraph of the Methods section.
